# Pathogen Discovery in the Post-COVID Era

**DOI:** 10.3390/pathogens13010051

**Published:** 2024-01-05

**Authors:** Cheng Guo, Jian-Yong Wu

**Affiliations:** 1Center for Infection and Immunity, Mailman School of Public Health, Columbia University, New York, NY 10032, USA; 2School of Public Health, Xinjiang Medical University, Urumqi 830017, China

**Keywords:** pathogen discovery, infectious disease, diagnosis

## Abstract

Pathogen discovery plays a crucial role in the fields of infectious diseases, clinical microbiology, and public health. During the past four years, the global response to the COVID-19 pandemic highlighted the importance of early and accurate identification of novel pathogens for effective management and prevention of outbreaks. The post-COVID era has ushered in a new phase of infectious disease research, marked by accelerated advancements in pathogen discovery. This review encapsulates the recent innovations and paradigm shifts that have reshaped the landscape of pathogen discovery in response to the COVID-19 pandemic. Primarily, we summarize the latest technology innovations, applications, and causation proving strategies that enable rapid and accurate pathogen discovery for both acute and historical infections. We also explored the significance and the latest trends and approaches being employed for effective implementation of pathogen discovery from various clinical and environmental samples. Furthermore, we emphasize the collaborative nature of the pandemic response, which has led to the establishment of global networks for pathogen discovery.

## 1. Introduction

In the past four years, the world has been hit from an unprecedented pandemic caused by the SARS-CoV-2. This global health crisis has resulted in immeasurable economic losses and millions of human deaths; however, its impact is anticipated to be long-lasting, affecting the world for years to come. As a matter of fact, almost all infectious disease outbreaks are attributed to newly emerging or reemerging pathogens, with more than 70 significant outbreaks since 1980 [1], which also include the 2003 SARS [2], 2009 Influenza H1N1 [3], 2013 Ebola virus [4], 2013 H7N9 avian influenza [5], 2015 Zika virus [6], and the 2018 and recent Nipah virus outbreaks [7]. Another yet ongoing Mpox outbreak was first declared as a multi-country outbreak by WHO in July 2022. As of today, in less than 17 months, there have been a total of 92,182 confirmed cases and 170 deaths reported across 113 countries. It is widely recognized now that the majority of these pathogens are zoonotic, originating from animal reservoirs. Factors including climate change, globalization, and the expansion of human populations into wildlife habitats have expedited the spread of previously barely or un-identified pathogens capable of causing epidemics. The repeatedly appeared outbreaks have become the top medical challenge and economic burden to the global public health system. Furthermore, there exists a vast array of chronic inflammatory diseases that are associated with infection, yet their microbiological aspects remain poorly elucidated [8,9].

As the countering measurement, the rapid and accurate detection or discovery of the pathogen is pivotal. It enables immediate tracking and monitoring of the spread of pathogens, understanding of the transmission dynamics of diseases, identifying potential reservoirs or vectors, and developing targeted interventions to control and prevent outbreaks. The complete procedure of pathogen discovery generally consists of the detection for microbial agents causing diseases and then the proof of causation relationship for leading diseases [10,11]. While under various circumstances, this concept of pathogen discovery can also be interpreted to diagnostics or identification of unknown pathogens in the clinical and environment settings [12].

The current COVID-19 pandemic is a crucial lesson on the importance of pathogen discovery. Both releases of SARS-CoV-2 genome by GISAID and virological.org kicked off a global surveillance effort [13]. Upon the information, the development of reliable diagnostic tests played a vital role in controlling the virus transmission. The development of BNT162b2 vaccine was initiated on 10 January 2020, right after the genome release by the Chinese Center for Disease Control and Prevention and GISAID. Multiple national institutions and research laboratories started their qPCR diagnostic development immediately after the sharing from virological.org by Zhang and Holmes. Though the pandemics were too unbeatable to contain, the swift discovery of the SARS-CoV-2 pathogen and its transparent public release allowed the maximal constrain of the devastating consequences. While the pandemic seems to be fading away, monitoring and surveillance of the transmission of SARS-CoV-2 with different variants will continue among human populations and wild animals in the coming years [14]. The demands of SARS-CoV-2 diagnosis, along with the supporting resources, have been and will continue to drive the development of pathogen discovery in return [15].

Collectively, prompt pathogen discovery does not only enable the timely measurements for minimizing the transmission but also contributes to the development of medical treatment and the creation of vaccines. In this perspective, we provide a summary of the recent innovations and paradigm shifts that have revolutionized the field of pathogen discovery in response to the COVID-19 pandemic. We also discuss the collaborative nature of the pandemic response, as the establishment and consensus of global collaborative networks have facilitated the sharing of data, standardization of protocols, and harmonization of bioinformatics pipelines, leading to the accelerated identification of emerging pathogens. From an optimistic standpoint, there are fair lessons to learn from the pandemic, and the world has started to realize the importance of embracing new technology and concepts in public health. By implementing these measurements and concepts, we can strengthen our preparedness to respond to future outbreaks, ultimately avoiding pandemics and saving lives.

## 2. Methods

To obtain comprehensive knowledge of the advancements in pathogen discovery prior to and during the COVID-19 pandemic, an extensive search was conducted in the PubMed (U.S. National Library of Medicine) database for relevant studies published up until November 2023. Various keywords, including “pathogen discovery,” “microbial detection,” “infectious disease diagnosis,” “nucleic acid-based method,” “cultivation-based method” “antigen diagnosis” “antibody diagnosis”, “causation of infectious disease,” and “Koch’s postulate”, were utilized either individually or in combination with similar search strategies described elsewhere [9]. Recent research articles and review publications were preferred, typically for those associated with SARS-CoV-2, but no limiting period was imposed in the screening. Additionally, books, general newspapers, and institutional websites were also reviewed to identify any potential sources for integration into the review manuscript.

## 3. Pathogen Detection Methods

### 3.1. Cultivation-Based Detection

With its earliest practices back in the middle of the nineteenth century, microbial cultivation has been long and widely used for isolation and identifying pathogens. Even for now, classification on microbial culture by examination of its gross morphological, macroscopic, physiology features are the benchmark for diagnosing numerous microorganisms in healthcare settings [16]. Cultivation-based methods offer several key benefits in terms of simplicity, reliability, and the absent need for advanced equipment; however, it exhibits limitations that include a time-consuming and laborious culturing procedure, susceptibility to contamination, and a dependence on phenotypic characterization. Moreover, cultivation can be applied to a mere 2~3% of the microbial population [17,18]. To address these limitations, cultivation methods have been improved in recent decades, with the emergence of innovative techniques for targeted or high-throughput cultivation. These advancements involve the use of diverse and specialized growth media components, precise control of environmental conditions, the utilization of heterogeneous host cells, and the incorporation of growth-promoting factors [19]. It is worth mentioning that the combination of microbial cultivation, typically for bacterial species, with matrix-assisted laser desorption ionization-time of flight mass spectrometry (MALDI-TOF) has proven to be a highly promising strategy for identifying various microbial species. The MALDI-TOF technique is characterized by its speed, sensitivity, and cost and labor efficiency. Nonetheless, the technology’s capacity for identifying novel pathogens is confined to instances where peptide mass fingerprints from relevant reference strains are available [20]. Given these contexts, it is concerning but perhaps not surprising that pathogen discovery continues to rely heavily upon cultivation-based methods. On 24 January 2020, China CDC isolated the first strain of the SARS-CoV-2 virus from the lower respiratory alveolar lavage fluid of a COVID-19 patient [21]. Subsequentially, the success of viral culturing has enabled the exploration of additional valuable prospects such as electron microscope images, studies on viral transmission and viability, and experiments involving cellular and animal models.

In contrast to cultivation, modernized methods of pathogen detection rely on identification of molecular signatures during infection. These molecular signatures come from either the microbial intrinsic traits or the host’s immune response to the pathogen. Depending on the types of molecular signatures, detection methods can be split into different categories, including nucleic acid-based, antigen-based, and immunology response-based detection methods (Figure 1). Additionally, it is worth mentioning that several alternative techniques exist, such as digital pathology, advanced imaging technologies, wearable biosensors, and pathogenic-microbiome analysis [22]. These methods have shown potential in aiding disease detection and diagnosis but are beyond the scope of this review.

### 3.2. Nucleic Acid-Based Detection

Nucleic acid-based methods generally involve the amplification of a specific signature of a nucleic acid sequence as its starting material is trivial to detect. When the nucleic acid is amplified with a certain detectable amount, the detection of nucleic acid can be achieved by fluorescent signaling, probe-based methods such as hybridization, direct visualization through electrophoresis, NGS sequencing, CRISPR, and others [23].

Polymerase Chain Reaction (PCR) is the most widely used method, which involves a series of heating and cooling cycles to separate the DNA strands, bind primers to the target sequence, and amplify the DNA fragment using DNA polymerase enzyme. Reverse Transcription Polymerase Chain Reaction (RT-PCR) was to first convert the RNA template into complementary DNA (cDNA) using reverse transcriptase enzyme, and then followed by regular PCR. The gold-standard diagnosis method of COVID-19 is RT-PCR, and trillions of real time RT-PCR tests have been carried out around the globe during the COVID-19 pandemic [24]. The use of COVID-19 qPCR testing directly enables early detection, contact tracing, identification of asymptomatic carriers, monitoring, and surveillance, mitigating cluster outbreaks and assessing the effectiveness of control measures.

In comparison to PCR-based methods, isothermal amplification technology offers efficient and specific DNA amplification under constant temperature conditions. This eliminates the need for complex equipment [25] and reduces the amplification time [23]. This technology also demonstrates greater tolerance towards impurities and inhibitors commonly found in field samples, ensuring high sensitivity and specificity without the necessity for purification or specialized handling. Moreover, isothermal amplification allows for amplification to be initiated using smaller amounts of target biomass but longer amplicons. These inherent advantages make isothermal amplification technology suitable for integration into portable biosensing devices, allowing improved sensitivity with on-site testing. However, isothermal amplification also encounters certain challenges like intricate design of primers, amplification that lacks specificity, high background signal, and the need for additional and costly enzymes and denaturing agents [26]. Collectively, the user-friendly nature of isothermal amplification technology makes itself an attractive candidate for point-of-care molecular tests. In the past four years, isothermal amplification market has gained huge recognition, and it is expected to grow at a compound annual growth rate of 12.2% during the next forecast decade. Popular isothermal amplification tests include RT-RPA (Recombinase Polymerase Amplification) [27], LAMP (Loop-Mediated Isothermal Amplification) [28], and NEAA (Nicking enzyme-assisted amplification) [29].

CRISPR-Cas-based detection relies on hybridization of the Cas protein with the DNA motif signatures of pathogens [30]. The hybridization event illustrates the presence of pathogenic DNA and generates a measurable signal, which can be fluorescence, electrochemical signal, colorimetric reaction, or a visual signal on a lateral flow device. CRISPR-coupled RT-RPA or RT-LAMP are advantageous alternatives to RT-PCR due to their cost-effectiveness and faster turnaround time. Currently, the lowest limit of detection (LOD) achieved among the CRISPR methods was two RNA copies per sample [31]. It has also been reported this test can be performed at a cost of less than USD 3.5, with a potential production scale cost as low as USD 0.7 [32]. The average accuracy of all CRISPR methods is approximately 96.5%, and the test results can be obtained in about 50 min [33]. Compared to RT-PCR, the CRISPR-based COVID test offers improved reliability and cost-effectiveness, making it a viable alternative. In combination with LAMP, CRISPR/cas-based assays have gained US FDA emergency use authorization during the COVID-19 pandemic.

Metagenomic Next Generation Sequencing (mNGS) is another technology that allows sequencing of DNA from various microbial species present in a sample simultaneously. The method involves extracting total DNA from the sample, followed by shotgun sequencing of DNA fragments that are assembled into contigs. These contigs can be then utilized to identify the presence of pathogens, microbial communities, and their genetic characteristics. In the past decade, mNGS has become an increasingly popular technique for diagnosing infections and characterizing microbial communities. When compared to alternative diagnostic technologies, one key advantage of mNGS is its unbiased sampling trait, allowing for the comprehensive identification of known pathogens, unexpected pathogens, and even the discovery of new organisms. The first evidence of SARS-CoV-2 was detected by metagenomics screening on unknown febrile patients from Wuhan, China, in late December 2019 before it came to public awareness. Another advantage is its ability to provide supplementary genomic information that supports evolutionary tracing, strain identification, and prediction of drug resistance. Furthermore, mNGS provide quantitative or semi-quantitative data by counting the number of sequenced reads, which is especially valuable for polymicrobial samples or cases where multiple pathogens are implicated in the disease process. It is also worth highlighting that the third generation of sequencing technology can provide high confidence and real-time surveillance during the COVID-19 pandemic, as demonstrated in numerous instances [34,35]. The global metagenomic sequencing market in terms of revenue was estimated to be worth USD 2.0 billion in 2023 and is poised to reach USD 4.5 billion in 2028, growing at a compound annual growth rate of 17.5% from 2023 to 2028.

A meta-analysis on the diagnostic test accuracy of mNGS indicates that sensitivity and specificity were 90% and 86% for blood, 75% and 96% for cerebrospinal fluid (CSF), and 84% and 67% for orthopedic samples, respectively [36]. However, the implementation of mNGS in clinical diagnostic laboratories has been limited, primarily due to operational complexity, cost considerations, and insufficient sensitivity compared to agent-specific PCR assays. Several strategies have been proposed to improve the sensitivity of mNGS, including the enrichment of pathogen templates through the removal of host nucleic acids using nuclease digestion and the depletion of ribosomal RNA [10]. Although these strategies have been helpful, none have achieved the sensitivity required for clinical applications. Aiming to enhance the sensitivity of mNGS and make it more suitable for clinical diagnostic purposes, a positive selection probe capture-based system was developed for the enrichment of targeted sequences [37,38,39]. Compared to the unbiased mNGS, capture-based sequencing can increase the sensitivity up to 1000-fold, reaching the detection sensitivity which is equivalent to RT-PCR [39]. A baited probe can be customized into different designs for specific usage. Among different laboratory developed tests, VirCapSeq has been first certified by the New York State Department of Health Clinical Laboratory Evaluation Program as viral pathogen detection for use in patient diagnosis and public health surveillance. When it comes to bacterial or fungal infections, where the species identification of pathogens can impact therapeutic decisions due to intrinsic resistance from antimicrobial resistance genes, the effect of antibiotics administration can be significantly improved through AMR detection associated with bacterial and fungal pathogens [40]. However, it is worth noting that capture-based sequencing has limitations in detecting novel pathogens that lack nucleotide sequence similarity represented in the probe-target pools, and its sequencing results may result in biased representation of sequences, reflecting variations in gene expression patterns and primer biases during library preparation.

### 3.3. Antigen-Based Detection

Aiming to provide quick and accessible testing options, the antigen-based detection is to identify the presence of antigens from the pathogen. Indeed, most self-tests or at-home tests are based on antigen detection. The COVID-19 pandemic also marked a milestone in public health as it witnessed the widespread adoption of self-testing by individuals, primarily with the SARS-CoV-2 antigen test. By empowering individuals to take an active role in their own testing, it enhances accessibility and convenience, leading to earlier detection, isolation, and treatment of infected individuals. Self-testing also reduces the burden on healthcare systems, allowing resources to be focused on those who require critical care. The two primary antigens targeted in SARS-CoV-2 detection assays are the spike protein (S protein) and the nucleocapsid protein (N protein) [41]. In the recent Cochrane review on various commercial SARS-CoV-2 antigen-detection rapid diagnostic tests (Ag-RDTs), it has been reported with a sensitivity of 56.2% (95% CI, 29.5% to 79.8%) and specificity of 99.5% (95% CI, 98.1% to 99.9%) [42]. The characteristic of Ag-RDTs determines that it can be effectively used in the first week after symptoms begin to confirm the infection. It is important to note that the effectiveness of Ag-RDTs in detecting infections may vary depending on the specific test used, the quality of the product, and the timing of testing in relation to symptom onset. While Ag-RDTs offer a valuable tool for early diagnosis during the first week of symptoms, confirmatory testing with other molecular methods, such as PCR, is still recommended in cases where Ag-RDT results are negative, but clinical suspicion remains high. This combined approach helps to enhance the accuracy of diagnosis and ensure that appropriate public health measures are implemented for effective control of the infection [43].

### 3.4. Antibody-Based Detection

Serological methods based on antibodies are used to target specific immunoglobulin isotypes (IgA, IgM, or IgG) or total antibodies in blood samples and occasionally in other body fluids like saliva. Serological diagnosis is essential when the genetic material (nucleic acid) of the infectious agent cannot be detected, at least 1 to 2 weeks after symptom onset for the host to develop immune responses. These assays offer valuable information about the humoral immune responses and the specific antigens they recognize, making them particularly useful for investigating complex chronic diseases and relevant molecular epidemiology research after outbreaks [11]. Known as ELISA, the earliest serological assay was first developed in 1971 [44]. Despite being considered labor-intensive and expensive, ELISA has been widely used in the clinical diagnosis and laboratory research. Regarding serological assays for SARS-CoV-2, the nucleocapsid (N) protein and the receptor-binding domain (RBD) of the S1 subunit of the S glycoprotein are the most frequently used antigens [45]. IgA and IgM have demonstrated advantages in detecting early immune responses, but concerns have been raised regarding their shorter lifespan compared to IgG in serum and saliva [46].

Antibody-based detection has also explored its potential in the high throughput form. One strategy is the use of a programmable microarray that consisted of a massive number of synthetic peptides. Depending on the material of the slide carrier, the array can accommodate from a range of from thousands to three million distinct linear peptides. This approach has greatly enhanced the assay sensitivity, enabling early diagnosis of diseases for Lyme disease, Acute Flaccid Myelitis, and others [47,48,49,50]. Another technique, VirScan, integrates high-throughput DNA oligo synthesis, bacteriophage display, and next-generation sequencing to achieve antibody profiling. By screening sera samples against a library of viral peptides, VirScan can identify the specific peptides recognized by antibodies through immunoprecipitation and sequencing [51]. VirScan has been successfully used to map the immune response to the SARS-CoV-2 virus in COVID-19 patients, providing insights into cross-reactivity and disease severity factors [52]. PepSeq is another multiplexed proteomic assay for pathogen discovery. Its protocol involves synthesizing the peptides of customizable targets of interest and linking them to cDNA tags in an in vitro and massively parallel manner. The resulting libraries enable highly multiplexed assays that utilize high-throughput sequencing to profile the binding or enzymatic specificities for targeted peptides [53].

## 4. Causation Relationship

It should be fully aware in pathogen discovery, detection and characterization of a microbial pathogen is merely the first step, which is followed by an establishment of the causal relationship (Figure 1). However, the proving of causation between the microbe and disease has been known as a challenging process due to its complexity involving a thorough investigation and evaluation of various factors. Developed by Robert Koch in the late 19th century, Koch’s postulates are a set of four criteria to guide this proving analysis. Koch’s postulates require that the microorganism must consistently be present in individuals suffering from the disease, isolated and grown in pure culture, cause the same disease when inoculated into a susceptible host, and then re-isolated from the experimentally infected host. Though Koch’s postulates have provided a valuable framework for establishing causality between a pathogen and a disease, limitations still exist, especially when studying viruses, asymptomatic infections, microbiomes, or chronic diseases. To compliment that, modifications have been proposed to the four original criteria by Rivers, Fredricks, Relman, and others over time (Figure 1) [54,55,56,57].

As of today, the fulfillment of Koch’s postulates or the modified postulates is considered the strongest evidence of causation. With the advent of new technology in pathogen detection, along with a broader scope of knowledge of human disease from different perspectives, it is acknowledged that the definitively proven cause of a disease is unlikely in many scenarios [54], unlike the causation between SARS-CoV-2 and COVID-19. Complementarily, numerous protocols can provide strong evidence for a potential causal relationship [57]. When classical hallmarks of infection are absent or when the mechanisms of pathogenesis are indirect or subtle, a statistical assessment of the strength of the epidemiological association can be employed using the presence of the agent or its molecular footprints (such as nucleic acid, antigen and antibody) [54]. By analyzing large groups of individuals, researchers can detect patterns and determine if there is a statistically significant difference to assess the relationship between a specific factor and the occurrence of a disease [58,59]. Experimental studies involve the manipulation of variables and can be used to assess causality as well, such as by examining the temporal relationship and dose–response relationship [60]. In a practical clinical setting, a timely call on the causation of disease is key for the containment of outbreaks in many cases. Even when sufficient but ambiguous evidence is formed, the measurement should be rapidly implemented as a response. It is also unnecessary for every criterion to be fulfilled to implicate a suspected microorganism in causing the disease. To summarize, when information from different perspectives was collected, the scientific consensus among experts in the field is crucial in establishing the cause of a disease. While expert evaluations and collective opinions based on available evidence can contribute to the credibility of a causal claim, they do not necessarily lead to a definitive conclusion. For instance, in cases similar to Torque teno viruses, additional cautions and experiment are always recommended to determine the true causation relationship between the microbe and disease [61].

## 5. The Impact of COVID Pandemic on the Field of Pathogen Discovery

The COVID-19 pandemic has imposed a profound effect on infectious diseases from both industrial and academic perspectives. In academia, the pandemic has strengthened the visibility of infectious disease specialists and scientists. This increased visibility is evident from the substantial increase in the number of scientific papers available, from 40,501 in 2019 to 58,099 in 2022, on PubMed database when searching for “infectious disease” (Figure 2). This reflects a growing emphasis on research and understanding of infectious diseases. In the industrial sector, the pandemic has caused disruptions in normal operations and economic growth while also driving a surge in the infectious disease and diagnostic industry. The global market for infectious disease diagnostics increased in size during the pandemic, doubling over the five-year period from 2017 to 2022 (Figure 2). Nonetheless, global exercise and applications of pathogen discovery is rapidly progressing driven by advancements in both industry and academia, leading to improved applications in clinical and environmental settings.

### 5.1. Rapid Pathogen Discovery from Clinical Samples

As the frontline of human infectious disease, rapid pathogen discovery in the clinical facility is critically important. Molecular-based techniques such as culturing, qPCR, sequencing, and ELISA are the primary methods utilized in healthcare settings, which optimize cost and resource efficiency. However, these approaches face certain limitations, including the need for expensive equipment and facility utilities, challenges in bioinformatics for data analysis, and the detection of antimicrobial resistance. To address these challenges, we discuss two emerging technologies from the COVID pandemic that have the potential to alleviate these difficulties: broader applications of point-of-care testing (POCT) [62] and the utilization of artificial intelligence (AI) [63,64].

The implementation of POCT has revolutionized rapid pathogen diagnosis by allowing medical diagnostic tests to be performed in close proximity to patients, rather than in a centralized laboratory. It facilitates immediate results, often within minutes, allowing healthcare providers to make accurate and timely diagnosis decisions. POCT devices play a crucial role in emergency scenarios or when prompt interventions are integral to the diagnosis process. Well-known POCT devices include the GeneXpert System (Cepheid), Liat Analyzer (Roche Diagnostics), Sofia Analyzer (Quidel), and FilmArray System (BioFire), which can be tailored to screen various infectious diseases. The demands of COVID-19 diagnostics over the past four years have accelerated the advancement of POCT. Notably, the US FDA has granted authorization for 37 at-home OTC COVID-19 diagnostic tests so far.

On the other hand, AI has taken a well-established yet ever-evolving position across various domains within the healthcare system in a relatively brief timeframe [65,66]. This encompasses analyses such as detecting outbreaks through the monitoring of internet traffic data, tracing contacts within these outbreaks, predicting variants, and analyzing transmission patterns from a public health standpoint. From a clinical medical perspective, AI contributes to automated image analysis, improved performance in clinical trials, and facilitates the interpretation of medical information on mobile devices by users. Specifically, the integration of machine learning algorithms empowers AI to automate processes, enhance diagnostic accuracy, and minimize the time needed for infectious disease diagnosis. For instance, through the analysis of existing datasets and the genetic or proteomic fingerprints for known pathogens and their variants, machine learning algorithms can learn patterns and features that distinguish them. TM-Vec and DeepBLAST are two deep learning techniques employed for aligning proteins with distantly related annotated proteins [67]. This application enables effective and accurate pathogen detection, harnessing the power of deep learning algorithms for protein alignments, ultimately leading to improved diagnostics. DeepAMR, by analyzing genetic information of *Mycobacterium tuberculosis* and resistance co-occurrence with deep denoising auto-encoder models, can determine the bacterial resistance profile, aiding in the selection of appropriate antibiotics [68]. In medical imaging, these algorithms can be trained to identify specific patterns or features associated with different pathogens [69]. Furthermore, machine learning algorithms assist in image recognition and, when combined with other clinical information, they possess great potential in supporting clinical decision making for infectious diseases, thereby enhancing healthcare outcomes [70].

### 5.2. Pathogen Discovery from Environmental Samples

The One Health concept recognizes the interconnectedness of human, animal, and environmental health, emphasizing the need for interdisciplinary collaboration among professionals in human health, veterinary medicine, environmental science, and related fields [71]. It is crucial to explore pathogen discovery in wild animals and environmental sources alongside clinical samples, as approximately 70% of human pathogens originate from zoonotic reservoirs. Pathogen spillover events occur when a reservoir population, usually asymptomatic carriers of the pathogen, encounters a new host population. This transmission can lead to outbreaks or even pandemics in the new host species [72]. To effectively understand and manage zoonotic diseases, it is essential to monitor reservoir species, such as animals or vectors, which carry and transmit these diseases. By monitoring these reservoirs, early signs of disease outbreaks can be detected, enabling the implementation of timely response strategies. Environmental surveillance plays a vital role in the early detection of emerging pathogens by systematically monitoring the environment for disease-causing agents before they cause outbreaks or epidemics. This surveillance includes sequencing and analyzing the genetic material from samples collected, including both the host organism, associated microorganisms, and environmental samples such as water [73], soil [74], or air [75].

During the Zika virus outbreak in Brazil in 2015, environmental samples like mosquitoes were collected and analyzed, leading to the identification of the virus, and facilitating a better understanding of its spread and the development of effective mosquito control strategies [76]. Similarly, in the case of Legionnaires’ disease, environmental samples from a hotel’s cooling tower were analyzed to identify the presence of *Legionella pneumophila*, aiding in pinpointing the source of the outbreak [77]. In the context of the ongoing COVID-19 pandemic, pathogen detection efforts have focused on nasal swabs, wastewater, and air samples to identify and track the presence and transmission of the SARS-CoV-2 virus [78,79,80]. Targeted genomic sequencing with probe capture was also applied for discovery and surveillance. Using a custom panel of 20,000 probes targeting all available bat coronavirus sequences, capture-based pathogen discovery and surveillance was conducted in bat swab specimens [81]. A median 6.1-fold improvement in read depth was achieved by probe capture, resulting in the identification of several novel alpha- and beta-coronaviruses in these specimens compared to conventional metagenomics. These examples underscore the importance of pathogen discovery from environmental samples in the control and prevention of infectious disease transmission. Hence, it is crucial to uphold and support surveillance practices such as monitoring sewage samples, migratory birds, wild animals, and high-risk animal workers. These measures should be continued to effectively mitigate the spread of infectious diseases.

## 6. Global Collaborative Networks and Data Sharing

Global collaborative networks and data sharing play a vital role in global pathogen discovery. These two components greatly contribute to our understanding of infectious diseases and our ability to detect, monitor, and respond to emerging pathogens. Global collaborative networks allow scientists and researchers from different regions to collaborate on identifying pathogens on the shared platform. On the other hand, sharing data on pathogen genetic sequences, clinical information, and outbreak locations allows for a faster response and the implementation of appropriate control measures.

The current COVID-19 pandemic response has indeed highlighted the collaborative nature of addressing global crises. Collaborating with scientists from different disciplines and regions allows for the sharing of best practices, innovative approaches, and lessons learned, especially when SAR-CoV-2 just started to spread across the globe in early 2020. Access to a diverse range of samples from different geographical locations is crucial for studying the genetic diversity of SAR-CoV-2, identifying new variants, and understanding its evolution. By sharing knowledge, training programs, and equipment, these networks help strengthen laboratory capacities, diagnostic capabilities, and research infrastructure globally. This ensures that all countries are equipped to detect and respond to infectious diseases effectively, as no country alone could contain the viral transmission by itself. Moreover, developing countries pose a significantly higher risk for pathogen emergence but with the least preparedness, therefore establishing infrastructure in these regions should be a priority in global pandemic prevention efforts. Towards protection and responding to microbial threats worldwide, GAPP, the Global Alliance for Preventing Pandemics, is one of the international collaborative centers that establishes sustainable infrastructure for infectious disease discovery, surveillance, and response through global capacity building [82]. It is also notable that, when integrated with infectious disease surveillance, the cloud computing also showed great potential in providing the information access and hardware resources around the globe for immediate data exchange and analysis [83]. COVID-19 reminds us that we are living in an interconnected global village with a common stake. All countries are closely connected, and we share a common future. By working together via collaborative networks and sharing data, we can rapidly and effectively respond to outbreaks of emerging pathogens [84].

## 7. Conclusions

During the COVID-19 pandemic, significant progress has been made in pathogen discovery. These advancements include global efforts in qPCR screening, widespread use of at-home antigen tests, the application of isothermal amplification, and the development of rapid genome sequencing for variant surveillance. The integration of metagenomics sequencing and artificial intelligence is expected to further enhance pathogen detection technologies. Furthermore, the adoption of the One Health concept, which recognizes the interconnectedness of human, animal, and environmental health will play a crucial role. Last but not the least, international collaboration will remain essential for sharing data, expertise, and resources, facilitating timely identification and response to emerging pathogens. Collectively, pathogen discovery will continue to be pivotal in post-COVID public health by enabling early detection and response, improving preparedness, enhancing surveillance systems, and implementing targeted public health interventions.

## Figures and Tables

**Figure 1 pathogens-13-00051-f001:**
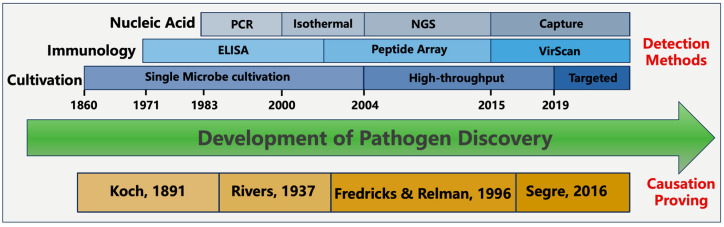
The development of pathogen discovery.

**Figure 2 pathogens-13-00051-f002:**
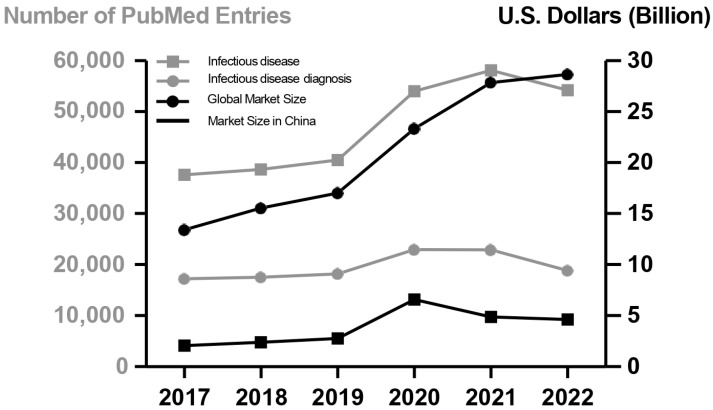
The COVID-19 pandemic results in a surge for infectious disease industry. The *x*-axis indicates the period from year 2017 to 2022, and the COVID-19 outbreak started in December 2019. The left *y*-axis (gray dots/lines) indicates the number of entries that can be found in NCBI PubMed database by searching the terms with either “infectious disease” or “infectious disease diagnosis”. The right *y*-axis (black dots/lines) indicates the evaluated global and regional (China) market size for infectious disease diagnosis. Relevant data from this figure were collected from the NCBI PubMed database, Precedence Research site, and Frost & Sullivan analytics report.

## Data Availability

No new data were created or analyzed in this study. Data sharing is not applicable to this article.

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
