# Peer review of "Pathogen Discovery in the Post-COVID Era"

_pathogens, 2024, doi:10.3390/pathogens13010051_

Round 1

Reviewer 1 Report

Comments and Suggestions for Authors

This is a potentially interesting review for non-specialists in virus detection field. Nonetheless, it prompts several questions and suggestions from my side.

L101 «Subsequentially, the success of viral culturing made it possible for the release of other valuable information, including electron microscope photographs, nucleic acid detection primers, and other various details.»

This example doesn't effectively illustrate the use of cultivation for obtaining primer structures. Primer acquisition is notably quicker using mNGS sequencing, suggesting a need to revise this statement for clarity.

L108 (and further in the text)

 «nucleotide acid-based» should be «nucleic acid based»

L126

The section on isothermal amplification is overly brief, lacking any detailed discussion of its fundamental principles and benefits, with the exception of analysis speed. A more comprehensive description would be beneficial.

L155

Metagenomic next-generation sequencing (mNGS) should be defined and briefly explained, considering its unfamiliarity to the general public, while bearing in mind that experts are unlikely to need such an overview.

L181

«Comparing to the unbiased mNGS, capture-based sequencing can increase the sensitivity up to 1000-fold which reach the detection sensitivity which is equivalent to qPCR»

Include the drawbacks as well, specifically that this method primarily enriches known or closely related viruses, but is much less effective in discovering entirely new ones.

L328

Artificial intelligence - Too brief; additional programs beyond those mentioned could be listed and briefly explained.

Reviewer 2 Report

Comments and Suggestions for Authors

This article titled"Pathogen discovery in the post-COVID era" discussed the important of pathogen discovery in the field of infectious disease, clinical microbiology and public health. However, I did not see much application description of technological innovations in the COVID-19 pandemic summarized here. The authors need to emphasize to write this part.

Comments on the Quality of English Language

English improvements are required.

Reviewer 3 Report

Comments and Suggestions for Authors

Reviewer 4 Report

Comments and Suggestions for Authors

The review by Guo and Wu has attempted to cover an essential aspect of summarizing the pathogen discovery aspect post-COVID-19 pandemic. The topic is of utmost importance to cover pathogen discovery's current and future perspectives.

“Pathogen Discovery in the Post-COVID Era” review broadly examines the multifaceted landscape of pathogen discovery in the post-COVID era, exploring recent techniques such as CRISPR-Cas-based detection, point-of-care testing (POCT), and AI computing. There are a few points that should be considered in this article before publication.

Following are the major corrections required for comprehensive and effective review:

1. Overall the review is very broadly written with minimal data. The authors should include more information for the reader. 

2. The current state of review covers several aspects of pathogen discovery vaguely and authors should try to make it concise with representations and tables. Authors should also try to make subheadings instead of long descriptions. 

3. A table representing various techniques with concise advantages, limitations, references, etc. can be incorporated for reader’s advantage.

4. In addition to methods already discussed, emerging techniques such as microfluidics, electrochemical biosensors, MALDI-TOF, and digital PCR will be integrated, providing a comprehensive insight into the evolving landscape of pathogen detection.

5. A section emphasizing the importance of pathogen discovery and surveillance in the context of the post-COVID-19 pandemic, elucidating the rationale behind deepening research efforts in this domain would be a significant addition.

6. Within "Pathogen Discovery from Environmental Samples" section, a refined discussion on environmental surveillance can be presented. This will include specific instances of global surveillance initiatives in the post-pandemic period, such as sewage monitoring, surveillance of migratory birds, zoo animal pathogen monitoring, etc. Additionally, data on major surveillance categories and publication trends from the past six years will be incorporated to offer a comprehensive understanding. Moreover, how pathogen emergence and discovery would be important in developing countries should be emphasized. 

7. One health concept is being worked out across the globe. This is a very important aspect and the author should have a separate section discussing the role of pathogen discovery in this regard.

8. A dedicated section on future perspectives, intending the trajectory of pathogen discovery research will provide valuable insights into the evolving landscape and encourage further technological advancement in this field.

Comments on the Quality of English Language

The English quality is average. It can be improved further by reducing the redundancy. The authors have written the review by generalizing multiple aspects. Similar concepts are written in different ways adding to the redundancy of the concepts.

Round 2

Reviewer 1 Report

Comments and Suggestions for Authors

Accept in present form

Reviewer 2 Report

Comments and Suggestions for Authors

good to go